# Development and Implementation of LED Street Lights with Bright and Extinguishable Controls and Power Converters

**DOI:** 10.3390/mi14071453

**Published:** 2023-07-20

**Authors:** Kai-Jun Pai, Liang-Hsun Wang, Ming-Hung Chen

**Affiliations:** 1Undergraduate Program of Vehicle and Energy Engineering, National Taiwan Normal University, Taipei City 10610, Taiwan; 2Department of Industrial Education, National Taiwan Normal University, Taipei City 10610, Taiwan; jimmy861127@gmail.com; 3Department of Electrical Engineering, Ming Chi University of Technology, New Taipei City 24301, Taiwan; mhchen@mail.mcut.edu.tw

**Keywords:** light-emitting diode, quasi-resonant (QR)-flyback converter, resonant valley

## Abstract

This study developed and implemented a driving power supply for light-emitting diode (LED) array streetlamps. The power stage was a quasi-resonant (QR)-flyback converter, its input power was the alternating-current power, and the LED array streetlamp was driven by the direct-current output power. The developed QR-flyback converter was operated in discontinuous conduction mode, and the pulse-width modulation (PWM) control chip was used to switch and conduct at the resonant valley of the drain-source voltage on the metal-oxide-semiconductor field-effect transistor (MOSFET) switch to reduce the switching loss. Moreover, the PWM control chip had a disable function, which was connected with a bright and extinguishable control circuit, and the high/low voltage level signal output by the Arduino development board can be used to control the output power of the QR-flyback converter, achieving bright and extinguishable controls for the LED array streetlamp.

## 1. Introduction

Street lighting is important equipment for sidewalks and roadways, and can impact traffic safety and the quality of the human environment to serve a sense of conformability and security. Moreover, street lighting can also improve the daytime and night appearance of the road environment. To ensure a good installation, street lighting standards require several performance indexes, such as illuminance, luminance, power qualities, and electrical conversion efficiencies [1,2,3].

The light-emitting diode (LED) can be used in indoor and outdoor environments including roadways, sidewalks, streets, building interiors, advertisement signboards, and ambient lighting. In street lighting applications, LEDs have more lifetime (50–100 k hours) compared with that of fluorescent or gas-discharge lamps, substantially reducing maintenance and replacement costs [4].

General LED drivers are composed of a power factor correction (PFC) circuit, DC–DC converter, and current control circuit. To promote the conversion efficiency and hardware reliability of the LED driver, the single-stage AC–DC converter as an alternative to the PFC circuit and the DC-DC converter has been developed and implemented [5].

Due to the greenhouse effect and climate change influences, science and technology development is placing more and more focus to energy saving and carbon reduction. Therefore, switching-mode power supplies (SMPSs) play a critical role in power conversion. SMPSs have isolated and non-isolated topologies. Non-isolated SMPSs include buck converters, boost converters, and boost-buck converters. Isolated SMPSs include full-bridge converters, half-bridge converters, forward converters, and flyback converters.

An isolated power converter separates the input alternating-current (AC) power from the output direct-current (DC) power by electrically and physically dividing the circuit into two sections, in order to prevent the AC power from influencing the load. The isolated AC–DC converter uses a high-frequency transformer to achieve galvanic isolation between the AC inlet and DC outlet.

Several benefits of isolated AC–DC converters include:Providing safety to humans and sensitive instruments against the high and potentially dangerous AC input source.Breaking ground loops.Avoiding floating outputs and voltage level shifting.

Therefore, isolated AC–DC converters have been used in medical, industrial, instrumentation, smart home, commercial electronic equipment, internet of things (IoT), telecommunication, battery charger, cell phone charger, vehicle or aircraft powertrains, military, and home applications [6].

Comparing the forward converter and the flyback converter, the transformer of the flyback converter dispenses with an additional demagnetization winding, hence the design difficulty and transformer winding cost can be reduced. Moreover, a quasi-resonant (QR)-flyback converter can achieve soft-switching for the power switch using the transformer’s primary inductance and the switch’s and circuit board’s parasitic capacitances; therefore, the conducted and radiated electromagnetic interferences can be reduced [7,8]. Otherwise, in order to improve the power conversion efficiency, retain the advantages of simple circuit configuration, and reduce the hardware cost, QR-flyback converters have become popular.

QR-flyback converters use the parasitic capacitance on the power switch and leakage inductance on the transformer to generate a resonant voltage when the power switch is turned off, and then the power switch can be turned on at the resonant voltage valley; therefore, soft-switching can be achieved to reduce switching losses, and electromagnetic interference can be effectively diminished. Moreover, the QR-flyback converter is an isolated SMPS because it possesses a transformer; furthermore, the QR-flyback converter can use a pulse-width modulation (PWM) control chip to correct the power factor of the input AC power; in summary, the QR-flyback converter is suitable as a driving power supply for LED array streetlamps [9,10]. Table 1 lists the characteristic differences between a hard-switching traditional flyback and soft-switching QR-flyback [7,8,11].

This study developed and implemented a QR-flyback converter driving an LED array streetlamp. Using a PWM control IC, the QR-flyback converter can achieve the power factor correction and drive the LED array streetlamp; moreover, the bright and extinguishable control circuit incorporating the PWM control IC could control the LED array streetlamp’s brightness and extinguishing operations.

## 2. Design Consideration of QR-Flyback Converter

The circuit block diagram of the QR-flyback converter is depicted in Figure 1, including the full-wave rectifier, input filter capacitor *C_in_*, snubber circuit, rectification and filter circuit, PWM control chip, *n*-channel metal-oxide-semiconductor field-effect transistor (MOSFET) switch Q, drain-source terminal capacitor *C_ds_*, bright and extinguishable control circuit, transformer T (including the magnetizing inductance *L_p_*, primary-side winding *n_pri_*, secondary-side winding *n_sec_*, and auxiliary winding *n_aux_*), diode D, and output filter capacitor *C_out_*; the input source is the AC voltage *v_ac_*, and the output power drives the load. The QR-flyback converter specification, transformer design, MOSFET switch specification, snubber design, secondary-side rectifier diode design, and input and output filter capacitor designs are described as follows:

### 2.1. QR-Flyback Converter Specification

The specifications of the developed QR-flyback converter are listed in Table 2.

### 2.2. Transformer Design

The terminal voltage across the transformer’s secondary side can be reflected to the transformer’s primary side, becoming a reflected voltage *V_R_*. The minimum peak value of the input AC voltage is *v_ac_*_(pk,min)_; a variable *kv* can be obtained and expressed as [12]:
*kv* = *v_ac_*_(pk,min)_/*V_R_*,(1)

Substitution of *V_R_* = 100 V and *v_ac_*_(pk,min)_ = 85$\sqrt{2}$ = 120 V into (1) can obtain *kv* = 1.2. Using the characteristic equation: *f*(*kv*) = (0.5 + *kv* × 1.4 × 10^−3^)/(1 + 0.82 × *kv*) [12], *f*(*kv*) = 0.25 can be obtained.

N27 and EF25 are the model numbers of the magnetic core material and transformer bobbin in the study application. The effective magnetic path length *l_e_* = 57.5 mm, effective magnetic cross-sectional area *A_e_* = 52.5 mm^2^, and effective volume *V_e_* = 3020 mm^3^ [13]. To ensure that the designed transformer is not operated in saturation, it is necessary to calculate the minimum magnetizing inductance *L_p_*_(min)_ in the primary-side winding of the transformer. *L_p_*_(min)_ can be expressed as [12]:(2)$$L_{p(\min)\;}=\frac{4{.3\;\times \;10}^{-6}{\;\times \;V}_{R}}{i_{p(\mathrm{pk})}\;\times \;0.93},$$
where *i_p_*_(pk)_ is the peak current passing the magnetizing inductance, which can be expressed as [12]:(3)$$i_{p(pk)}=\frac{{2\times P}_{in(\max)}}{v_{ac(\mathrm{pk},\min)}\times f(kv)},$$
where *P_in_*_(max)_ is the maximum input power. In Table 1, the output power of the QR-flyback converter is *P_out_* = 52.5 W, and the conversion efficiency is set at 80%, hence *P_in_*_(max)_ can be calculated as 65.63 W; in this study, *P_in_*_(max)_ = 70 W was used. Substitution of *P_in_*_(max)_ = 70 W, *v_ac_*_(pk,min)_ = 120 V, and *f*(*kv*) = 0.25 into (3) can yield *i_p_*_(pk)_ = 4.7 A. Substitution of the aforementioned parameters and *V_R_* = 100 V into (2) can obtain *L_p_*_(min)_ = 99 μH. According to [14], another magnetizing inductance equation can be expressed as:(4)$$L_{p}=\frac{v_{ac(\mathrm{pk},\min)}}{\left({1+k}_{v}\right){\times f}_{sw(\min)}{\times i}_{p(\mathrm{pk})}},$$
where *f_sw_*_(min)_ is the minimum operating frequency of the MOSFET switch, and its value is *f_sw_*_(min)_ = 80 kHz. Substitution of *v_ac_*_(pk,min)_ = 120 V, *kv* = 1.2, *f_sw_*_(min)_ = 80 kHz, and *i_p_*_(pk)_ = 4.7 A into (4) can yield *L_p_* = 145.07 μH, which is greater than *L_p_*_(min)_ = 99 μH.

The calculating expression of *n_pri_* can be expressed as [12,14]:(5)$$n_{pri}=\frac{L_{p}{\times i}_{p(\mathrm{pk})}{\times 10}^{3}}{B_{max}{\times A}_{e}},$$

Substitution of *L_p_* = 145.07 μH, *i_p_*_(pk)_ = 4.7 A, *B_max_* = 0.3 mT, and *A_e_* = 52.5 mm^2^ into (5) can obtain *n_pri_* = 43.29 ≅ 44, because the winding turns in practical applications are the positive integer.

The turn ratio *n* of transformer can be expressed as [12]:(6)$$n=\frac{V_{R}}{{(V}_{out}{+V}_{f})}=\frac{n_{pri}}{n_{sec}},$$
where *V_f_* is the forward bias voltage of D. Substitution of *V_f_* = 0.8 V, *V_R_* = 100 V, and *V_out_* = 35 V into (6) can obtain *n_sec_* = 15.75 ≅ 16. Using both *n_pri_* = 44 and *n_sec_* = 16, *n* = 2.75 can be obtained.

The calculating expression of *n_aux_* can be expressed as [12]:(7)$$n_{aux}=\frac{V_{aux}{\times n}_{sec}}{V_{out}},$$
where *V_aux_* is a voltage across the auxiliary winding; in this study, the operating power of the PWM control chip was set to 15 V. Substitution of *V_aux_* = 15 V, *V_out_* = 35 V, and *n_sec_* = 16 into (7) can obtain *n_aux_* = 6.86 ≅ 7. Aforementioned parameters include *n* = 2.75, *n_pri_* = 44, *n_sec_* = 16, *n_aux_* = 7, and *L_p_* = 145.07 μH.

### 2.3. Metal-Oxide-Semiconductor Field-Effect Transistor (MOSFET) Specification

The withstand voltage and current are the important specifications for MOSFET switch selection. When the MOSFET switch is turned off, the leakage inductance on the transformer and paratactic capacitance on the MOSFET switch cause the resonant voltage spike *v_spike_*, hence the withstand voltage of the MOSFET switch must be greater than *v_spike_*, which can be expressed as [15]:(8)$$v_{spike}=i_{p(\mathrm{pk})}\sqrt{\frac{L_{leak}}{C_{ds}}},$$
where *L_leak_* is the leakage inductance on the primary side of the transformer. It is usually 1% to 3% of *L_p_*, hence *L_leak_* = *L_p_* × 1% = 145.07 μH × 1% = 1.45 μH in this study. Moreover, *C_ds_* is the drain-source terminal capacitance on the MOSFET switch, and the *C_ds_* = 470 pF was used in this study. Substitution of *i_p_*_(*p*k)_ = 4.7 A, *L_leak_* = 1.45 μH, and *C_ds_* = 470 pF into (8) can obtain *v_spike_* = 261.06 V. The withstand voltage of the MOSFET switch can be expressed as [12]:*V_break_* = *v_ac_*_(pk,max)_ + *V_R_* + *v_spike_*.(9)

Substitution of *v_ac_*_(*p*k,max)_ = 198 V, *V_R_* = 100 V, and *v_spike_* = 261.06 V into (9) can obtain *V_break_* = 559.06 V, hence the withstand voltage of the MOSFET switch must be greater than 559.06 V. Moreover, the withstand current of the MOSFET switch must be greater than *i_p_*_(pk)_ = 4.7 A. Furthermore, the small *C_ds_* and gate terminal charge *Q_g_* can be selected for switching loss reduction. The model number STF10N80K5 of the MOSFET switch [16] was used in this study, its specifications listed in Table 3.

### 2.4. Snubber Circuit

The resonant voltage spike generated by the QR-flyback converter exceeds the withstand voltage of the MOSFET switch, resulting in device damage. The voltage spike can be reduced by the snubber circuit. The snubber circuit elements include a resistor *R_snub_*, *C_snub_*, and *D_snu_*, as shown in Figure 2. *R_snub_* and *C_snub_* can be expressed as [17]:(10)$$C_{snub\;}=\frac{L_{leak}{\times i}_{p(pk)}^{2}}{v_{spike}{\times (v}_{spike}{+2V}_{R})},$$
(11)$$R_{snub}\geq \frac{1}{f_{sw(\min)}{\times C}_{snub}\times \ln\left(1+\frac{v_{spike}}{V_{R}}\right)},$$

Substitution of *L_leak_* = 1.45 μH, *i_p_*_(pk)_ = 4.7 A, *v_spike_* = 261.06 V, and *V_R_* = 100 V into (10) can obtain *C_snub_* = 266.11 pF. Substitution of the aforementioned parameters and *f_sw_*_(min)_ = 80 kHz into (11) can yield *R_snub_* = 36.59 kΩ. The diode D_snu_ of the snubber circuit can use a fast recovery diode, whose recovery time can reduce the switching loss of *D_snu_*.

### 2.5. Rectification Diode

The withstand voltage calculation of the rectification diode D can be expressed as [12]:(12)$$V_{d}{=V}_{out}{+v}_{\mathrm{ac}(\mathrm{pk},\max)}\times \frac{n_{\sec}}{n_{\mathrm{pri}}}.$$

Substitution of *V_out_* = 35 V, *v_ac_*_(pk,max)_ = 198 V, *n_pri_* = 44, and *n_sec_* = 16 into (12) can obtain *V_d_* = 107 V.

According to the transformer reflection law, the peak current calculation of the D on the secondary-side of the transformer can be expressed as:
*i_sec_*_(pk)_ = *i_p_*_(pk)_ × *n*. (13)

Substitution of *n* = 2.75 and *i_p_*_(pk)_ = 4.7 A into (13) can obtain *i_sec_*_(pk)_ = 12.93 A ≅ 13 A. Therefore, the withstand voltage and current of the D must be selected that are greater than 107 V and 13 A, respectively.

### 2.6. Output Filter Capacitor

The filter capacitor can be used to stabilize the output voltage of the SMPS. When the load was changed, the current ripple magnitude was related to the equivalent series resistor (ESR) of the filter capacitor [18]; the low ESR can reduce the current ripple when the load changes. The output filter capacitor calculation can be expressed as [19]:(14)$$C_{\mathrm{out}}\geq \frac{I_{\mathrm{out}}\times \mathrm{ncp}}{f_{\mathrm{sw}(\min)}\times \Delta V_{\mathrm{out}}},$$
where △*V_ou__t_* is the peak-to-peak value of the output voltage, the and *ncp* is the number of the internal clock cycle for the PWM control chip needed by the control loop to reduce the duty cycle from maximum to minimum value. The *ncp* can be set at 10 to 20 [19].

Substitution of △*V_out_* = *V_out_* × 1% = 35 × 1% = 0.35 V, *I_out_* = 1.5 A, *f_sw_*_(min)_ = 80 kHz, and *ncp* = 20 into (14) can obtain *C_out_* = 1071.43 μF.

Because the ESR of the single electrolytic capacitor was of high value, this study used the two electrolytic capacitors of 680 μF and the ceramic capacitor of 220 pF in parallel connection to reduce the ESR of *C_out_*. Due to the fact that the capacitance value (680 μF + 680 μF + 220 pF) was higher than the calculating result (1071.43 μF), the influence of the capacitance value error in the practical application can also be eliminated.

### 2.7. Input Filter Capacitor

When the QR-flyback converter is used as a DC–DC converter, the input AC power can be filtered by a capacitor after passing through a full-wave rectifier to obtain an input DC voltage. However, input DC voltage has a voltage ripple *v_bk_*_(ripple)_, as shown in Figure 3. In Figure 3, the cycle ratio *D_bulk_* = *t*_1_/*t*_2_ during the input filter capacitor charging is set to 0.2 [20].

Moreover, the voltage ripple on the input filter capacitor was set to the maximum input AC voltage of 10% (*v_ac_*_(max)_ = 140$\sqrt{2}$ × 10% ≅ 20 V); therefore, the minimum voltage across the filter capacitor *v_bk_*_(min)_ = *v_ac_*_(pk,min)_ − 20 V = 120 V − 20 V = 100 V. Furthermore, according to [21], the minimum voltage of the input filter capacitor can be expressed as:(15)$$v_{\mathrm{bk}(\min)}=\sqrt{{2\times v}_{\mathrm{ac}(\mathrm{pk},\min)}^{2}-\frac{P_{\mathrm{in}}\times (1-D_{\mathrm{bulk}})}{C_{\mathrm{in}}{\times f}_{\mathrm{line}}}},$$
which can be written as:
(16)$$C_{\mathrm{in}}=\frac{P_{\mathrm{in}(\max)}\times (1-D_{\mathrm{bulk}})}{{(2\times v}_{\mathrm{ac}(\mathrm{pk},\min)}^{2}-v_{\mathrm{bk}(\min)}^{2}{)\times f}_{\mathrm{line}}},$$
where *f_line_* is the frequency of the input AC power source. Substitution of *v_bk_*_(min)_ = 100 V, *D_bulk_* = 0.2,*P_in_*_(max)_ = 70 W, *v_ac_*_(pk,min)_ = 120 V, and *f_line_* = 60 Hz into (16) can obtain *C_in_* = 49.65 μF. However, the QR-flyback converter in this study was used as an AC/DC converter, and *C_in_* was used as a high-frequency filter; therefore, *C_in_* can choose a capacitance value 200 times smaller than 49.65 μF [16]. In this study, the *C_in_* = 220 nF was chosen with the withstand voltage of 630 V (this withstand voltage was greater than *v_ac_*_(*p*k,max)_ = 198 V) in the practical application.

## 3. Experimental Results

In this study, the experimental voltage and current measurements are shown in Figure 1, including input AC voltage *v_ac_*, input AC current *i_in_*, full-wave rectification voltage *v_bk_*, transformer secondary-side current *i_sec_*, output voltage *V_out_*, output current *I_out_*, MOSFET drain-source voltage *v_ds_*, MOSFET gate source-voltage *v_gs_*, and control voltage *V_ctrl_*.

To confirm that the QR-flyback converter can output the rated voltage and current under the input AC condition, the peak value of *v_bk_* was 155 V (110 V_rms_), *V_out_* = 35 V, and *I_out_* = 1.5 A, as shown in Figure 4.

Under the full-load operation and *v_ac_* = 110 V_rms_, *v_ac_* and *i_in_* measurement waveforms are shown in Figure 5. Waveforms of *v_ac_* and *i_in_* were in-phase, which verified that the QR-flyback converter designed in this study achieved the power factor correction.

The power analyzer (PW3390, HIOKI E.E. Corp., Nagano, Japan) was used to measure the harmonic distortion rate. Under the full-load operation and *v_ac_* = 110 V_rms_, the fifth-order harmonic histogram and harmonic record table are shown in Figure 6. In Figure 6a, the voltage, current, and power generated the maximum harmonic in the first (1st)-order; the current produced odd harmonics above the third (3rd)-order, whose values were low. Figure 6b recorded that the THD percentage was 0.06%, which addressed the IEC 61000-3-2 Class-C standard [22].

Under the full load operation and *v_ac_* = 85 V_rms_, the experiment and simulation waveforms of *v_gs_*, *v_ds_* and *i_sec_* are shown in Figure 7. The operating frequencies of *v_gs_* were 76.92 kHz (experiment) and 78.13 kHz (simulation), and the highest voltages of *v_ds_* were 300 V (experiment) and 300 V (simulation), respectively. *v_gs_* was changed at the resonant valley of *v_ds_*, and the MOSFET switch was turned on. The PSIM software was used for the simulation. Moreover, the peak currents of *i_sec_* were 18 A (experiment) and 17 A (simulation), respectively; the result of *i_sec_* based on (13) was 13 A.

Under the full load operation and *v_ac_* = 140 V_rms_, the experiment and simulation waveforms of *v_gs_*, *v_ds_* and *i_sec_* are shown in Figure 8. The operating frequencies of *v_gs_* were 97.1 kHz (experiment) and 97 kHz (simulation), respectively; the highest voltages of *v_ds_* were 390 V (experiment) and 380 V (simulation), respectively. *v_gs_* was changed at the resonant valley of *v_ds_*, and the MOSFET switch was turned on. Moreover, the peak currents of *i_sec_* were 16 A (experiment) and 17 A (simulation), respectively.

The external control signal *V_ext_* (Figure 3) was used to control the *V_ctrl_* (Figure 3) voltage level of the PWM control chip, and then *V_out_* of the QR-flyback converter can be controlled, as shown in Figure 9. In Figure 9a, when *V_ext_* = 0, *V_ctrl_* = 2 V, and *V_out_* = 35 V, this experiment represented the fact that the bright and extinguishable control circuit enabled the QR-flyback converter to drive the LED array streetlamp at *V_out_* = 35 V; therefore, the LED array streetlamp could be lighted. In Figure 9b, when *V_ext_* = 5 V, *V_ctrl_* = 0, and *V_out_* = 0, this experiment represented the fact that the bright and extinguishable control circuit disabled the output voltage of the QR-flyback converter; therefore, the LED array streetlamp was dimmed.

At the different *v_ac_* (85 to 140 V_rms_), *I_out_* was changed from 0.1 to 1.5 A, the efficiency measurements were recorded in Figure 10. The minimum efficiency was about 32% under the *v_ac_* = 140 V_rms_ and *I_out_* = 0.1 A; the maximum efficiency was about 85% under the *v_ac_* = 140 V_rms_ and *I_out_* = 1.5 A.

*V_ext_* (Figure 1 and Figure 9) was generated by the Arduino development board and combined with the QR-flyback converter to drive the LED array streetlamp system, as shown in Figure 11. In Figure 11, the three LED array streetlamps were controlled achieving bright and extinguishable operations at different times, when the model car moved to different positions.

The implement block diagram of the LED array streetlamp is depicted in Figure 12, its operation described as follows:The QR-flyback converter was started up.The external signal *V_ext_* was detected to control the bright and extinguishable control circuit.When *V_ext_* was a low voltage level, the LED array streetlamp employed the bright operation; when *V_ext_* was a high voltage level, the LED array streetlamp employed the extinguishable operation.

## 4. Conclusions

This study developed a driving power supply for LED array streetlamps. The driving power supply was a QR-flyback converter, which combined with a PWM control chip to achieve the power factor correction of the input AC power, and output to drive the LED array streetlamps. This study provided detailed design considerations in the parameter calculations of the power stage devices and verified their correctness with experimental results. The Arduino development board was combined with the QR-flyback converter to drive the LED array streetlamp system.

## Figures and Tables

**Figure 1 micromachines-14-01453-f001:**
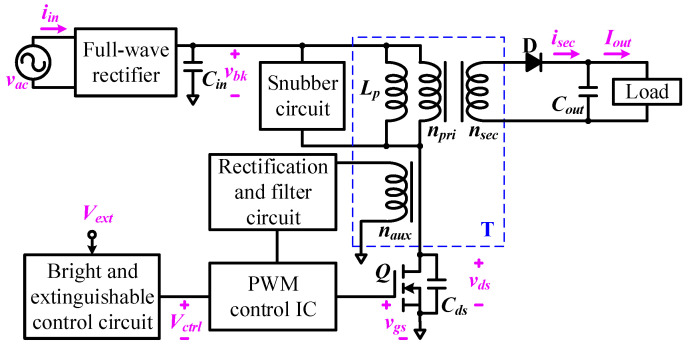
Circuit block diagram of QR-flyback converter.

**Figure 2 micromachines-14-01453-f002:**
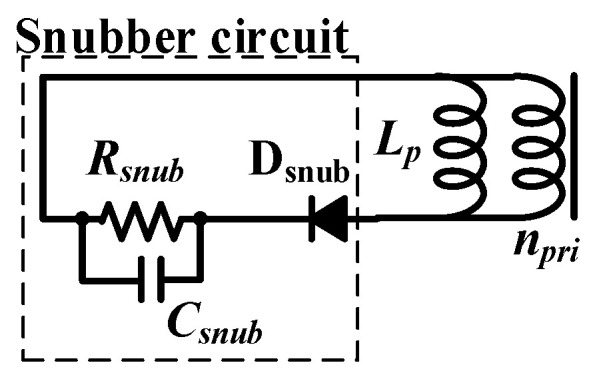
Snubber circuit.

**Figure 3 micromachines-14-01453-f003:**
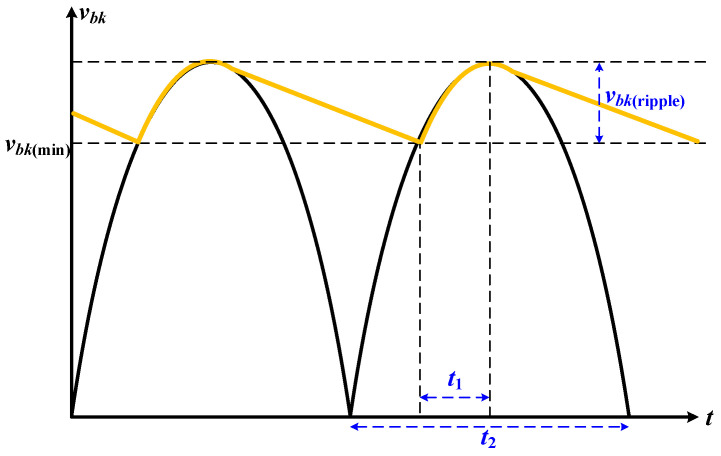
Input voltage ripple on *C_in_*.

**Figure 4 micromachines-14-01453-f004:**
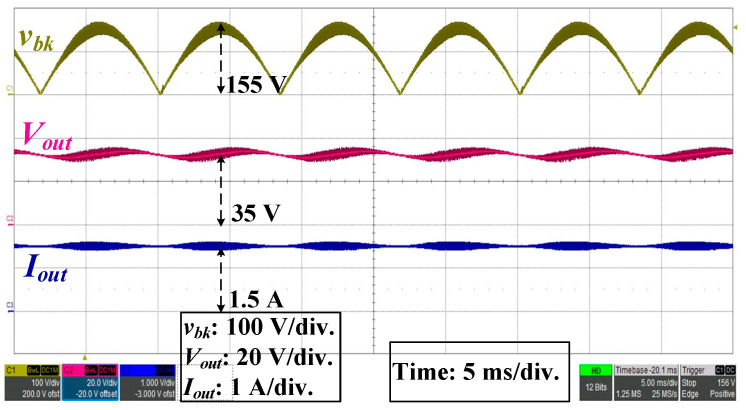
Waveforms of full-wave rectification, output voltage and current.

**Figure 5 micromachines-14-01453-f005:**
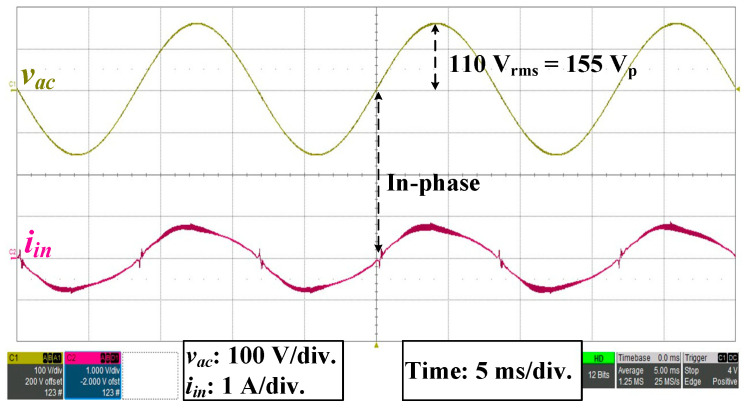
*v_ac_* and *i_in_* were in-phase.

**Figure 6 micromachines-14-01453-f006:**
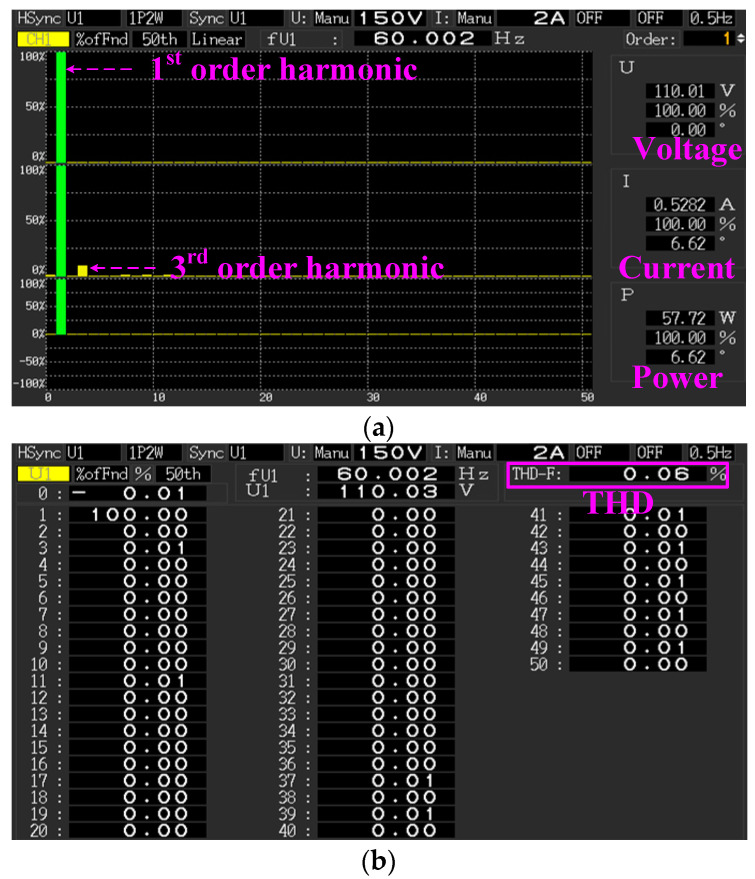
Power analyzer measured harmonic distortion: (**a**) Harmonic histogram; (**b**) Harmonic record table.

**Figure 7 micromachines-14-01453-f007:**
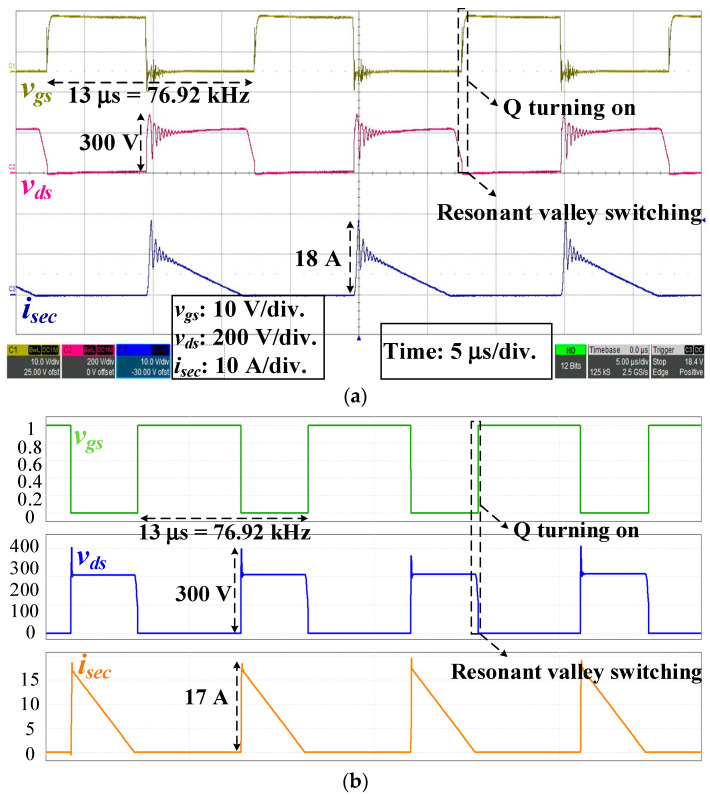
*v_gs_*, *v_ds_*, and *i_sec_* at the AC input of 85 V_rms_: (**a**) Experiment; (**b**) Simulation waveforms.

**Figure 8 micromachines-14-01453-f008:**
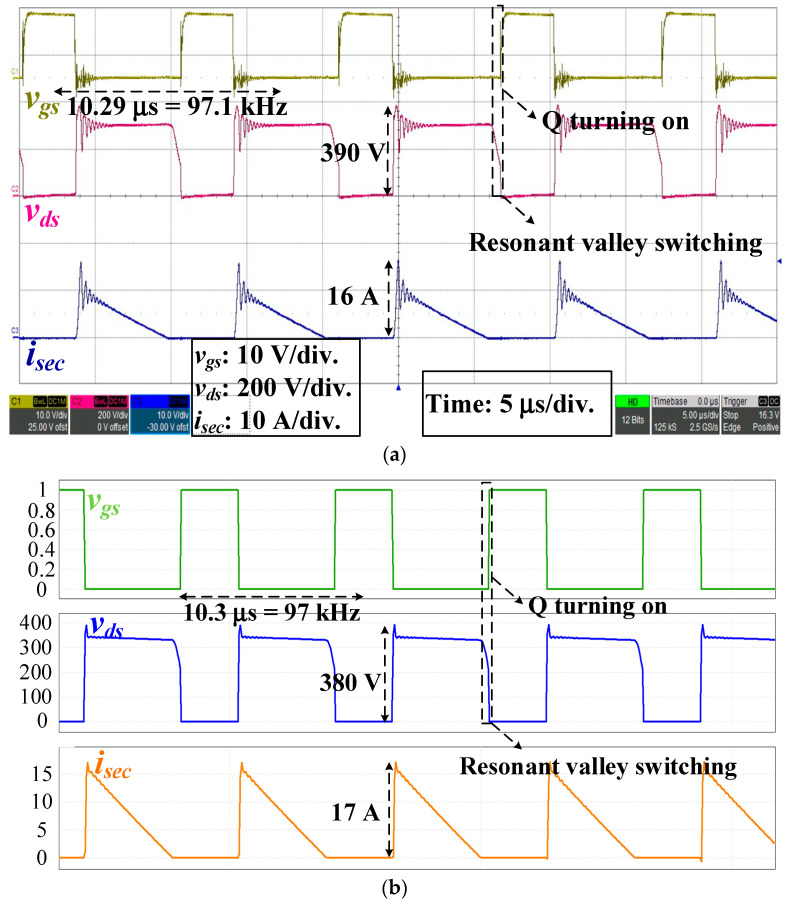
*v_gs_*, *v_ds_*, and *i_sec_* at the AC input of 140 V_rms_: (**a**) Experiment; (**b**) Simulation waveforms.

**Figure 9 micromachines-14-01453-f009:**
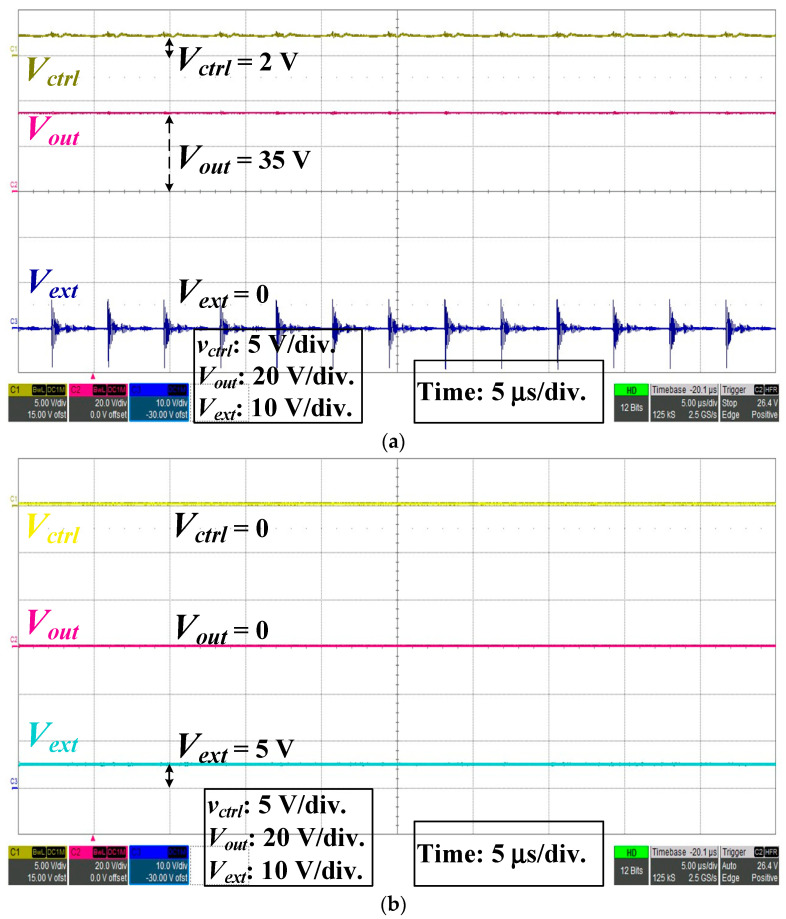
*V_ext_*, *V_ctrl_*, and *V_out_* measurements of bright and extinguishable control circuit: (**a**) Light bright; (**b**) Light extinguish.

**Figure 10 micromachines-14-01453-f010:**
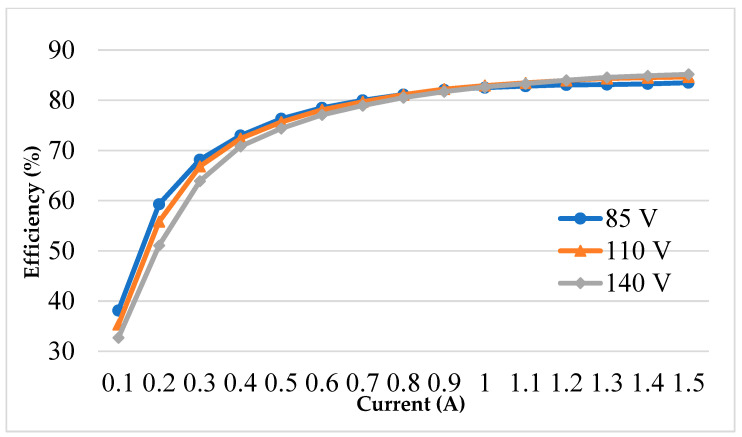
Efficiency measurement.

**Figure 11 micromachines-14-01453-f011:**
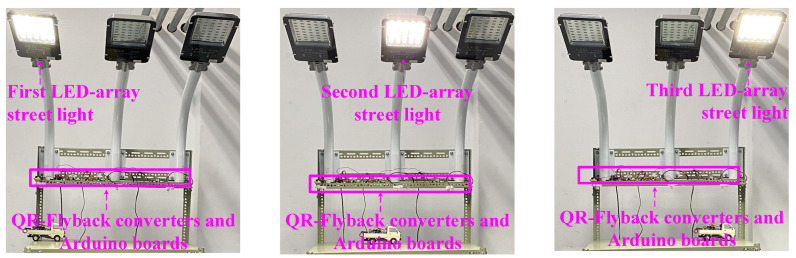
Arduino development board combined with QR-flyback converter to drive LED array streetlamp.

**Figure 12 micromachines-14-01453-f012:**
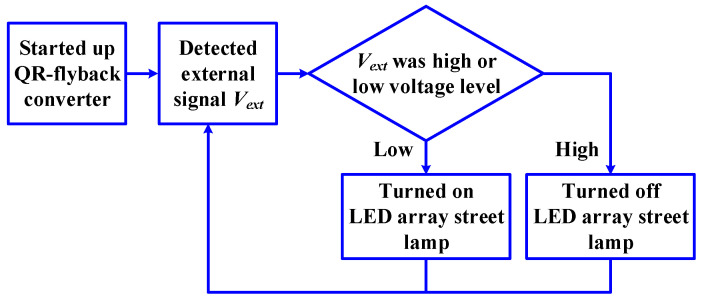
Implement block diagram of the LED array streetlamp.

**Table 1 micromachines-14-01453-t001:** Characteristic differences between a hard-switching traditional flyback and soft-switching QR-flyback.

	Traditional Flyback in the Discontinuous Conduction Mode Operation	QR-Flyback
Conduction losses of power switch and output rectification diode	High	High
Reversed recovery loss of output rectification diode	Low	Low
Switching loss of power switch	Low	Low
Output filter capacitance	Large	Large
Feedback and stability designs	Simple	General
Switching frequency	Constant	Adjustable
Average efficiency	Low	High

**Table 2 micromachines-14-01453-t002:** QR-flyback converter specification.

Description and Notation	Specification
Input AC voltage (*v_ac_*)	85 to 140 V_rms_
Output voltage (*V_out_*)	35 V
Output current (*I_out_*)	1.5 A
Output power (*P_out_*)	52.5 W
Maximum duty cycle ratio (*D_max_*)	0.47
Efficiency	>80%

**Table 3 micromachines-14-01453-t003:** MOSFET switch (STF10N80K5) specification [16].

Description	Specification
Gate terminal charge	22 nC
Gate-drain terminal charge	5.5 nC
Gate-source terminal charge	13.2 nC
Input paratactic capacitance	74 pF
Output paratactic capacitance	20 pF
Withstand voltage	800 V
Withstand current	9 A
Turn-on resistance	0.6 Ω

## Data Availability

Data sharing is not applicable.

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
