# Peer review of "Development and Implementation of LED Street Lights with Bright and Extinguishable Controls and Power Converters"

_micromachines, 2023, doi:10.3390/mi14071453_

Round 1
Reviewer 1 Report
The study reported development of a QR-flyback converter for LED array street lamps with PWM chip and MOSFET switch to perform light control. As-designed lamps show good performance. The authors need to address the following questions before considering publication.
(1) Some words in e.g. Figure 4-5 are too small to recognize.
(2) Introduction is short and did not provide sufficient background of the study with nearly no reference.
(3) The design aims to improve efficiency and reduce cost. The authors could provide comparison of the power conversion efficiency and hardware cost with classical flyback converters in a table to show the advantage of the design.
(4) The authors could provide power efficiency and flicker at different dimming level and discuss the results.
There are quite a few typos/grammar mistakes in the text. The authors need to read through the text and check carefully.
Author Response
Response to Reviewer Comments
Reviewer 1:
The study reported development of a QR-flyback converter for LED array street lamps with PWM chip and MOSFET switch to perform light control. As-designed lamps show good performance. The authors need to address the following questions before considering publication.
Response: Thank you for considering my manuscript and for your positive evaluation.
- Some words in e.g. Figure 4-5 are too small to recognize.
Response: Please review pages 7 and 8 of the new manuscript.
Numeric texts have been enlarged, as shown in Figures 4 and 5.
- Introduction is short and did not provide sufficient background of the study with nearly no reference.
Response: Please review pages 1 and 2 of the new manuscript.
Introduction section contexts and references have been increased.
An isolated power converter separates the input alternating-current (AC) power from the output direct-current (DC) power by electrically and physically dividing the circuit into two sections, in order to prevent the AC power from influencing the load. The isolated AC-DC converter uses a high-frequency transformer to achieve the galvanic isolation between the AC inlet and DC outlet.
Several benefits of isolated AC-DC converters include:
- Providing safety to humans and sensitive instruments from the high and potentially dangerous AC input source.
- Breaking ground loops.
- Avoiding floating outputs and voltage level shifting.
Therefore, isolated AC-DC converters have been used in medical, industrial, instrumentation, smart home, commercial electronic equipment, internet of things (IoT), telecommunication, battery charger, cell phone charger, vehicle or aircraft powertrains, military, and home applications [1].
Moreover, a quasi-resonant (QR)-flyback converter can achieve the soft-switching for the power switch using the transformer’s primary inductance and the switch’s and circuit board’s parasitic capacitances; therefore, the conducted and radiated electromagnetic interferences can be reduced [2,3]. Otherwise, in order to improve the power conversion efficiency, retain the advantages of simple circuit configuration, and reduce the hardware cost, the QR-flyback converters have been used in popularity.
Table 1 lists the characteristic differences between a hard-switching traditional flyback and soft-switching QR-flyback [2,3,6].
|
|
Traditional flyback in the discontinuous conduction mode operation |
QR-flyback |
|
Conduction losses of power switch and output rectification diode |
High |
High |
|
Reversed recovery loss of output rectification diode |
Low |
Low |
|
Switching loss of power switch |
Low |
Low |
|
Output filter capacitance |
Large |
Large |
|
Feedback and stability designs |
Simple |
General |
|
Switching frequency |
Constant |
Adjustable |
|
Average efficiency |
Low |
High |
This study developed and implemented a QR-flyback converter driving the LED array street lamp. Using a PWM control IC, the QR-flyback converter can achieve the power factor correction and drive the LED array street lamp; moreover, the bright and extinguishable control circuit incorporating the PWM control IC could control the LED array street lamp for its bright and extinguishable operations.
- The design aims to improve efficiency and reduce cost. The authors could provide comparison of the power conversion efficiency and hardware cost with classical flyback converters in a table to show the advantage of the design.
Response: Please review page 2 of the new manuscript.
Table 1 lists the characteristic differences between a hard-switching traditional flyback and soft-switching QR-flyback [2,3,6].
Table 1. characteristic differences between a hard-switching traditional flyback and soft-switching QR-flyback.
|
|
Traditional flyback in the discontinuous conduction mode operation |
QR-flyback |
|
Conduction losses of power switch and output rectification diode |
High |
High |
|
Reversed recovery loss of output rectification diode |
Low |
Low |
|
Switching loss of power switch |
Low |
Low |
|
Output filter capacitance |
Large |
Large |
|
Feedback and stability designs |
Simple |
General |
|
Switching frequency |
Constant |
Adjustable |
|
Average efficiency |
Low |
High |
- The authors could provide power efficiency and flicker at different dimming level and discuss the results.
Response: Please review page 11 of the new manuscript.
At the different vac (85 to 140 Vrms), Iout was changed from 0.1 to 1.5 A, the efficiency measurements were recorded in in Figure 10. The minimum efficiency was about 32% under the vac = 140 Vrms and Iout = 0.1 A; The maximum efficiency was about 85% under the vac = 140 Vrms and Iout = 1.5 A.
Vext (Figures 1 and 9) was generated by the Arduino development board and combined with the QR-flyback converter to drive the LED array street lamp system, as shown in Figure 11. In Figure 11, the three LED array street lamps were controlled achieving bright and extinguishable operations at different times, when the model car moved to different positions.

Reviewer 2 Report
The motivation of the study and state-of-the-art work in "LED Street Lights with 2 Bright and Extinguishable Controls and Power Converters" is not mentioned.
Introduction is too short.
Can we use non-isolated converter for this application? If not, why.
Add simulation results.
Add references for all design equations.
Draw implementation block diagram.
fine
Author Response
Reviewer 2:
The motivation of the study and state-of-the-art work in "LED Street Lights with 2 Bright and Extinguishable Controls and Power Converters" is not mentioned.
Response: Thank you for considering my manuscript and for your positive evaluation.
Introduction is too short.
Response: Please review pages 1 and 2 of the new manuscript.
Introduction section contexts and references have been increased.
An isolated power converter separates the input alternating-current (AC) power from the output direct-current (DC) power by electrically and physically dividing the circuit into two sections, in order to prevent the AC power from influencing the load. The isolated AC-DC converter uses a high-frequency transformer to achieve the galvanic isolation between the AC inlet and DC outlet.
Several benefits of isolated AC-DC converters include:
- Providing safety to humans and sensitive instruments from the high and potentially dangerous AC input source.
- Breaking ground loops.
- Avoiding floating outputs and voltage level shifting.
Therefore, isolated AC-DC converters have been used in medical, industrial, instrumentation, smart home, commercial electronic equipment, internet of things (IoT), telecommunication, battery charger, cell phone charger, vehicle or aircraft powertrains, military, and home applications [1].
Moreover, a quasi-resonant (QR)-flyback converter can achieve the soft-switching for the power switch using the transformer’s primary inductance and the switch’s and circuit board’s parasitic capacitances; therefore, the conducted and radiated electromagnetic interferences can be reduced [2,3]. Otherwise, in order to improve the power conversion efficiency, retain the advantages of simple circuit configuration, and reduce the hardware cost, the QR-flyback converters have been used in popularity.
Table 1 lists the characteristic differences between a hard-switching traditional flyback and soft-switching QR-flyback [2,3,6].
Table 1. characteristic differences between a hard-switching traditional flyback and soft-switching QR-flyback.
|
|
Traditional flyback in the discontinuous conduction mode operation |
QR-flyback |
|
Conduction losses of power switch and output rectification diode |
High |
High |
|
Reversed recovery loss of output rectification diode |
Low |
Low |
|
Switching loss of power switch |
Low |
Low |
|
Output filter capacitance |
Large |
Large |
|
Feedback and stability designs |
Simple |
General |
|
Switching frequency |
Constant |
Adjustable |
|
Average efficiency |
Low |
High |
This study developed and implemented a QR-flyback converter driving the LED array street lamp. Using a PWM control IC, the QR-flyback converter can achieve the power factor correction and drive the LED array street lamp; moreover, the bright and extinguishable control circuit incorporating the PWM control IC could control the LED array street lamp for its bright and extinguishable operations.
Can we use non-isolated converter for this application? If not, why.
Response: Please review page 1 of the new manuscript.
An isolated power converter separates the input alternating-current (AC) power from the output direct-current (DC) power by electrically and physically dividing the circuit into two sections, in order to prevent the AC power from influencing the load. The isolated AC-DC converter uses a high-frequency transformer to achieve the galvanic isolation between the AC inlet and DC outlet.
Several benefits of isolated AC-DC converters include:
- Providing safety to humans and sensitive instruments from the high and potentially dangerous AC input source.
- Breaking ground loops.
- Avoiding floating outputs and voltage level shifting.
Therefore, isolated AC-DC converters have been used in medical, industrial, instrumentation, smart home, commercial electronic equipment, internet of things (IoT), telecommunication, battery charger, cell phone charger, vehicle or aircraft powertrains, military, and home applications [1].
Add simulation results.
Response: Please review pages 8 to 10 of the new manuscript.
Under the full load operation and vac = 85 Vrms, the experiment and simulation waveforms of vgs、vds and isec are shown in Figure 7. The operating frequencies of vgs were 76.92 kHz (experiment) and 78.13 kHz (simulation), and the highest voltages of vds were 300 V (experiment) and were 300 V (simulation), respectively. vgs was changed at the resonant valley of vds, the MOSFET switch was turned on. The PSIM software was used for the simulation. Moreover, the peak currents of isec were 18 A (experiment) and 17A (simulation), respectively; the calculating result of isec based on (13) was 13 A.
Under the full load operation and vac = 140 Vrms, the experiment and simulation waveforms of vgs、vds and isec are shown in Figure 8. The operating frequencies of vgs were 97.1 kHz (experiment) and 97 kHz (simulation), respectively; the highest voltages of vds were 390 V (experiment) and were 380 V (simulation), respectively. vgs was changed at the resonant valley of vds, the MOSFET switch was turned on. Moreover, the peak currents of isec were 16 A (experiment) and 17A (simulation), respectively.
Add references for all design equations.
Response: The new manuscript has increased citations to all equations.
Draw implementation block diagram.
Response: Please review pages 11 and 12 of the new manuscript.
The implement block diagram of the LED array street lamp is depicted in Figure 12, its operation describes as follows:
- The QR-flyback converter was started up.
- The external signal Vext was detected to control the bright and extinguishable control circuit.
- When Vext was low voltage level, the LED array street lamp was the bright operation; when Vext was high voltage level, the LED array street lamp was the extinguishable operation.

Round 2
Reviewer 1 Report
The questions have been well addressed and the manuscript have been improved.
N/A
Author Response
- The questions have been well addressed and the manuscript has been improved.
Response: Many thanks for your positive evaluation.
Reviewer 2 Report
introduction is still short
Author Response
- Introduction is still short.
Response: Please review page 1 of the new manuscript.
Introduction section contexts and references have been increased.